# CRISPR/nCas9-Based Genome Editing on GM2 Gangliosidoses Fibroblasts via Non-Viral Vectors

**DOI:** 10.3390/ijms231810672

**Published:** 2022-09-14

**Authors:** Andrés Felipe Leal, Javier Cifuentes, Valentina Quezada, Eliana Benincore-Flórez, Juan Carlos Cruz, Luis Humberto Reyes, Angela Johana Espejo-Mojica, Carlos Javier Alméciga-Díaz

**Affiliations:** 1Institute for the Study of Inborn Errors of Metabolism, Faculty of Science, Pontificia Universidad Javeriana, Bogotá 110231, Colombia; 2Department of Biomedical Engineering, Universidad de los Andes, Bogotá 111711, Colombia; 3Grupo de Diseño de Productos y Procesos (GDPP), Department of Chemical and Food Engineering, Universidad de los Andes, Bogotá 111711, Colombia

**Keywords:** CRISPR/nCas9, genome editing, GM2 gangliosidoses, non-viral vectors, Sandhoff, Tay–Sachs

## Abstract

The gangliosidoses GM2 are a group of pathologies mainly affecting the central nervous system due to the impaired GM2 ganglioside degradation inside the lysosome. Under physiological conditions, GM2 ganglioside is catabolized by the β-hexosaminidase A in a GM2 activator protein-dependent mechanism. In contrast, uncharged substrates such as globosides and some glycosaminoglycans can be hydrolyzed by the β-hexosaminidase B. Monogenic mutations on *HEXA*, *HEXB*, or *GM2A* genes arise in the Tay–Sachs (TSD), Sandhoff (SD), and AB variant diseases, respectively. In this work, we validated a CRISPR/Cas9-based gene editing strategy that relies on a Cas9 nickase (nCas9) as a potential approach for treating GM2 gangliosidoses using in vitro models for TSD and SD. The nCas9 contains a mutation in the catalytic RuvC domain but maintains the active HNH domain, which reduces potential off-target effects. Liposomes (LPs)- and novel magnetoliposomes (MLPs)-based vectors were used to deliver the CRISPR/nCas9 system. When LPs were used as a vector, positive outcomes were observed for the β-hexosaminidase activity, glycosaminoglycans levels, lysosome mass, and oxidative stress. In the case of MLPs, a high cytocompatibility and transfection ratio was observed, with a slight increase in the β-hexosaminidase activity and significant oxidative stress recovery in both TSD and SD cells. These results show the remarkable potential of CRISPR/nCas9 as a new alternative for treating GM2 gangliosidoses, as well as the superior performance of non-viral vectors in enhancing the potency of this therapeutic approach.

## 1. Introduction

The GM2 gangliosidoses are a group of three different diseases known as Tay–Sachs (TSD), Sandhoff (SD), and the AB variant, caused by mutations on the *HEXA*, *HEXB* of *GM2A* genes, respectively [1,2]. The *HEXA* and *HEXB* genes encode for the alpha (α) and beta (β) subunits, respectively [2], while the *GM2A* gene encodes for the GM2 activator protein (GM2AP) [3,4]. Under physiological conditions, the α and β subunits can dimerize, giving rise to two isoforms denominated β-hexosaminidase A (Hex A, αβ) or B (Hex B, ββ). Moreover, the αα dimerization leads to the release of the β-hexosaminidase S (Hex S) [5].

The lack of or reduced β-hexosaminidase activity, as well as absence of a functional GM2AP, results in an impaired GM2 ganglioside degradation inside the lysosome. Patients suffering from GM2 gangliosidoses generally face a significant central nervous system (CNS) impairment [2], although visceromegaly may take place in SD as well [6,7]. Neuroinflammation-related neurodegeneration seems to be responsible for the CNS compromise in patients [2,8,9]. According to the severity of the disease, life expectancy can be as low as two years [8,9].

Despite this poor prognosis, no therapies are currently approved for GM2 gangliosidoses. Several clinical trials are ongoing for some therapeutic alternatives based on enzyme replacement [10,11], pharmacological chaperones [12,13], substrate reduction [14], and hematopoietic stem cell transplantation [2]. However, their full potential is yet to be corroborated. Regarding gene therapy (GT), classical strategies that rely on viral vectors for delivery have been tested recently with promising results [15,16]. However, efforts to optimize them for implementation in clinical practice are ongoing [2,17].

Since its implementation as a biotechnological tool, the clustered regularly interspaced short palindromic repeats/CRISPR-associated protein 9 (CRISPR/Cas9) system has opened a new horizon for gene editing due to its capacity to rewrite the genome in any genomic region with relative ease [18,19]. In fact, clinical trials are underway for treating several pathologies involving cancer, infectious diseases, and rare conditions [20]. Although the CRISPR/Cas9 system was previously tested in SD mice by Ou et al. (2020), the donor plasmid included the presence of the chimeric β-hexosaminidase Hex M [21]. This heterodimeric protein is formed by the presence of the catalytic motif of the α subunit and the stable interface of the β subunit, which is required for the interaction with the GM2 ganglioside [22]. Despite the positive results reported for this strategy, the presence of a chimeric protein in humans may induce a marked immune response [23], which may affect therapeutic efficacy.

Recently, we reported on the use of a CRISPR/Cas9-based genome editing approach that relied on the Cas9 nickase (D10A), demonstrating a high on-target effect and non-detectable off-target effects [24,25]. In this study, we evaluated this CRISPR/nCas9-based gene-editing system to induce the insertion of normal *HEXA* and *HEXB* cDNA into the *AAVS1* locus on in vitro models of TSD and SD, respectively. After initial validation experiments, we extended the findings to long-term experiments using two delivery strategies based on both classical lipo-transfection and novel magnetolipo-transfection methods.

## 2. Results

### 2.1. Molecular Validation of the CRISPR/nCas9 System on HEK293 Cells

#### CRISPR/nCas9 Leads the Integration of Expression Cassettes Containing *HEXA* or *HEXB* cDNA into the *AAVS1* Locus

Previously, it was demonstrated that the CRISPR/nCas9 system described in this study allowed the knock-in of the Donor AAVS1:GALNS [24]. To confirm that the CRISPR/nCas9 system may integrate the Donor AAVS1:HEXA and Donor:HEXB into the locus *AAVS1*, we performed a homologous recombination (HR) experiment. Using a PCR with primers flanking a region of the *AAVS1* locus (upstream of the left homologous recombination arm, HRA-L) and the CMV promoter sequence (Figure 1a), a PCR product around 1.5 kb was observed in cells transfected with CRISPR/nCas9 and Donor AAVS1:HEXA or Donor AAVS1:HEXB (Figure 1b). The PCR failed to amplify any target when the cells were transfected only with Donor plasmids, suggesting that HR-independent integration did not occur. As expected, Sanger sequencing of the PCR products revealed CMV upstream of the HRA-L (Figure 1c). Likewise, the presence of the mutated PAM was observed in the HRA-L to avoid the Cas9 cut once the HR takes place [24].

After transfection of the CRISPR/nCas9 and donor plasmids, the expression of mCherry and EGFP was corroborated by fluorescence microscopy and flow cytometry (Figure 1d,e). The transfection ratios for double-positive cells of 40.1 ± 10.9%, 30 ± 8.4%, and 17.05 ± 6.3% were detected for the Donor AAVS1:Empty, Donor AAVS1:HEXA, or Donor AAVS1:HEXB, respectively (Figure 1e). Regarding the transcription assays, a significant increase was observed in HEK293 cells transfected with Donor AAVS1:HEXA (Fold-change: 975 ± 97.4, *p* ≤ 0.0001) and Donor AAVS1:HEXB (Fold-change: 298 ± 134.7, *p* = 0.0049), compared to the untreated controls (Figure 1f). Consistently, the β-hexosaminidase activity was increased after Donor AAVS1:HEXA or Donor AAVS1:HEXB transfection, reaching 534.9 ± 0.7 U/mg (Figure 1g) and 13.4 ± 0.1 U/mg (Figure 1h) for Hex A and Hex B, respectively. These activity levels were significantly higher than those of untreated HEK293 cells (Hex A: 498 ± 1.9 U/mg, Hex B: 6.2 ± 0.6 U/mg). Taken together, these results suggest the effective HR-dependent insertion of cassettes containing the wild-type versions of *HEXA* and *HEXB* genes into the *AAVS1* locus. Moreover, they provide evidence for the expression of α and β active sub-units capable of dimerization and subsequent biological activity against artificial substrates.

### 2.2. CRISPR/nCas9-Based Genome Editing on GM2 Fibroblasts

#### 2.2.1. Stable Genome Edition Is Achieved on TSD and SD Fibroblasts upon CRISPR/nCas9-Based Treatment

To test the efficacy of the CRISPR/nCas9-based system on in vitro models of TSD and SD, we followed the β-hexosaminidase activity up to 7 and 15 days (Figure 2) in fibroblasts using an LPs-mediated CRISPR/nCas9 delivery. For TSD fibroblasts, the Hex A activity increased significantly after 7 (18.1 ± 3.7% of WT levels, *p* = 0.0029) and 15 (10.8 ± 1.2% of WT levels, *p ≤* 0.0001) days post-edition (Figure 2a). Likewise, for SD fibroblasts the Hex B activity increased significantly after 7 (13.1 ± 2.7% of WT levels, *p* = 0.0015) and 15 (11.1 ± 1.6% of WT levels, *p* = 0.0019) days post-treatment (Figure 2b). In transfected TDS fibroblasts, we did not detect an increase in extracellular Hex A activity, while a modest increase in the Hex B activity was observed in the supernatant from SD fibroblasts at both 7 (Hex B: 4.3 ± 2.5% of WT levels, *p* = 0.0476) and 15 (Hex B: 6.9 ± 2.7% of WT levels, *p* = 0.0335) days post-transfection (Figure 2b). Transfection of Donor AAVS1 carrying *HEXA* or *HEXB* cDNA in the absence of the CRISPR/nCas9 plasmid failed to reach any significant enzyme activity 7 and 15 days after treatment. Overall, the β-hexosaminidase enzyme activity assays suggest successful short- and medium-term genome editing following a single transfection of the CRISPR/nCas9 system.

#### 2.2.2. MLPs Increase the Transfection of CRISPR/nCas9-Related Plasmids with Non-Cytotoxic Effects in SD and TSD Fibroblasts

The non-viral vectors showed average hydrodynamic diameters of 126 nm, 199 nm, 211 nm, and 273 nm for MNPs, MNPs@Ag-pD/BUF-II, lecithin-based liposomes, and MLPs, respectively (Figure 3a). Regarding the Z-potential, we found a value of −29.8 ± 0.6 mV for MNP, which becomes more positive when the MNPs@Ag-pD/BUF-II nanobioconjugates (−9.9 ± 0.9 mV) were synthesized. For empty liposomes, Z-potential resulted in −30 ± 1 mV, while after forming the MNPs@Ag-pD/BUF-II nanobioconjugates, this value decreased to −42.8 ± 1.1 mV (Figure 3a). Finally, the DNA loading capacity showed a linear trend up to 2000 ng, with a maximum capacity that was calculated as 35 ng/µg of MNPs@Ag-pD/BUF-II (Figure 3b).

We found that incubation at 37 °C with Rho-MLPs led to successful internalization of nanobioconjugates. On the other hand, at 4 °C, the MLPs showed a peripherical distribution around the plasma membrane (Figure 3c). Before conducting long-term edition assays, we conducted cell viability assays in SD and TSD fibroblasts with the MLPs-based vectors. Interestingly, TSD fibroblasts were more susceptible to MLPs-mediated apoptosis (~11%) than SD fibroblasts (~8%) (Figure 3d). However, DIC analysis failed to detect an evident change in cellular morphology (Figure 3e). This difference was also observed for LDH assays but not for the MTT ones (Appendix A). Finally, MLPs showed an improved transfection ratio compared to LPs, in both TSD (6.7% vs. 52.6%, respectively) and SD (4.6% vs. 29.8%, respectively) fibroblasts (Figure 3f).

#### 2.2.3. LPs-Assisted CRISPR/nCas9 Delivery Elicits a Better Response for Long-Term Edition Than MLPs on TSD and SD Fibroblasts

Despite the significant transfection efficiency increase observed with MLPs (see above results), we did not observe a consistent increase in the mRNA levels (Figure 4) or β-hexosaminidase activity (Figure 5) one-month post-transfection using MLPs. For instance, a slight increase in the mRNA levels, compared to untreated fibroblast, was achieved for TSD (fold change: 0.07 ± 0.01, *p* = 0.0102) and SD fibroblasts (fold change: 0.09 ± 0.007, *p* = 0.0009). Nonetheless, the LPs-assisted delivery led to reaching wild-type levels in the transcription assays for both TSD (fold change: 1.2 ± 0.3) and SD (fold change: 1.5 ± 0.5) fibroblasts (Figure 4).

Consistently, the intracellular Hex A (10.1 ± 2% of WT levels, *p* = 0.001) and Hex B (11.9 ± 1.2% of WT levels, *p* ≤ 0.0001) activity was higher with the LPs-assisted delivery compared to the MLPs, for which Hex A and Hex B activity levels were 1.9 ± 0.4% (*p* = 0.0229) and 2.2 ± 0.5% (*p* = 0.0051) of WT levels, respectively (Figure 5a). Surprisingly, extracellular Hex A activity levels reached 10% of wild-type levels when MLPs were employed for delivery of the CRISPR/nCas9 system. Still, it remains undetectable after the LPs-assisted delivery (Figure 5b). In contrast, ~50% of wild-type extracellular Hex B activity was observed for LPs, while an increase of 11% was achieved with MLPs (Figure 5b). Taken together, these results demonstrated the capacity of our CRISPR/nCas9-based genome editing strategy for long-term edition; however, using the MLPs as a non-viral vector resulted in a more attenuated outcome with respect to LPs. Further optimization of the MLPs is likely to will allow us to improve the genome editing ratio.

#### 2.2.4. Lysosomal Storage Significantly Decreases in TSD and SD Fibroblasts after LPs-Assisted CRISPR/nCas9 System Delivery

Although primary accumulation in SD and TSD corresponds to GM2 gangliosides inside the lysosome, secondary GAGs accumulation can also occur [2,26,27]. To evaluate the global impact of our CRISPR/nCas9-based genome editing strategy on this classical biomarker, we determined the total GAGs accumulation and the lysosomal mass. Untreated TSD and SD fibroblasts showed an increased GAGs amount with respect to the wild type, suggesting a global impairment in the lysosomal function. In concordance with the β-hexosaminidase activity assays for short-, medium-, and long-term edition, a significant reduction in the GAGs, close to wild-type levels, was observed by using LPs-assisted delivery in TSD and SD fibroblasts (Figure 6a). A substantial decrease of 40% and 27% in lysosomal mass in TSD and SD fibroblasts was observed after LPs-assisted delivery, respectively (Figure 6b,c). On the other hand, we failed to detect any significant changes in the GAGs or lysosomal mass for TSD and SD fibroblasts with MLPs-assisted delivery (Figure 6). These experiments showed the need to reach high intracellular β-hexosaminidase activity to recover the lysosomal mass, which was possible with the LPs but not with the MLPs. Further assays such as lipids staining with NileRed [10,28] will be conducted in future work to support these observations, where the Donor AAVS1 will be employed without GFP, given that fluorescence from both reporters should be collected with an FITC-associated laser [10].

#### 2.2.5. Global Oxidative Stress Is Positively Impacted by CRISPR/nCas9-Based Genome Editing

Since oxidative stress has been described as a critical factor in the GM2 gangliosidoses pathology [2], we evaluated the mtROS levels and the presence of NO-derived species on TSD and SD fibroblasts. As expected, untreated GM2 fibroblasts showed higher mtROS levels than wild-type. Surprisingly, despite the previous findings for enzyme activity and GAGs storage, a positive outcome in the mtROS levels was observed with both LPs- and MLPs-assisted delivery. In this regard, a higher reduction for LPs (53% compared to untreated cells) compared with MLPs (32% compared to untreated cells) was observed for TSD fibroblasts (Figure 7). Similar behavior was found for SD fibroblasts with LPs- (30% compared to untreated cells) and MLPs- (~24% compared to untreated cells) assisted delivery (Figure 7).

No changes were observed in the NO-derived species, neither in TSD nor SD fibroblasts, suggesting that the CRISPR/nCas9 system and the MLPs do not increase NO-related oxidative stress (Appendix A). Taken together, these results suggest that CRISPR/nCas9-based genome edition positively impacted the oxidative profile of TSD and SD fibroblasts.

## 3. Discussion

Recently, we developed a gene therapy strategy based on a CRISPR/nCas9 system with encouraging findings in MPS IVA fibroblasts [24,25]. In this work, we took advantage of this genome editing tool to evaluate its usefulness in GM2 gangliosides fibroblasts. In this new approach, after a double enzyme restriction with the *Mfe*I and *Mlu*I enzymes, we introduced the encoding *HEXA* or *HEXB* cDNA into Donor AAVS1:Empty, resulting in an expression cassette that was further inserted into the *AAVS1* locus in a double-nicking Cas9-dependent cut (Figure 1a–c). These results show that several genes can be easily cloned in our Donor AAVS1 through enzymatic restriction, supporting its use as a versatile donor plasmid for treating different genetic diseases.

In the transcription assays, HEK293 cells transfected with Donor AAVS1:HEXA showed higher mRNA levels than those transfected with Donor AAVS1:HEXB. However, the enzyme assays resulted in higher Hex B activity than that of Hex A, suggesting that ββ homodimers could be more stable than αβ heterodimers, as reported previously [29,30]. In fact, under physiological conditions, it has been postulated that HEXB remains as a lower transcript gene to favor the αβ dimerization while avoiding the overproduction of ββ homodimers [29]. Despite this, we cannot discard the presence of Hex S, which results from the αα dimerization and can hydrolyze MUGS [31]. Further experiments to elucidate the expression level of Hex S will be conducted in the future.

When LPs were used as a vector, we observed that the CRISPR/nCas9 system might lead to a stable increase in Hex A and Hex B activities after one month of treatment, reaching levels >10% of wild-type levels (Figure 5a). It has been proposed that more than 10% of normal Hex A and Hex B activity should be achieved to allow the GM2 ganglioside degradation [7,32]. Moreover, for Hex B, we detected ~50% of wild-type levels in the extracellular media, supporting our previous hypothesis about the dimerization of the beta-beta subunits. These findings contrast with our results for MPS IVA, in which up to 40% of wild-type GALNS activity levels were achieved [24]. Since, in both studies, we employed the same CRISPR/nCas9-based system, except for the cDNA, we hypothesize that the difference in restored levels may be due, at least in part, to the exoproteolytic trimming required by the β-hexosaminidases in the lysosome [33,34,35] which is absent for the GALNS enzyme. In this regard, GALNS requires activation by the formylglycine-generating enzyme at the Cys79 during its processing within the endoplasmic reticulum [36,37].

Despite the above, our results for Hex A and Hex B activity agree with the findings for CRISPR/Cas9 and IDUA, where up to 10% of normal activity was reported after a long-term assay of MPS I fibroblasts using non-viral vectors [38,39]. Interestingly, using CRISPR/nCas9, we have found higher Hex A activity with respect to classical GT assisted by viral vectors. For instance, Flotte et al. (2022) reported an increase in the Hex A activity as low as 0.9% of wild-type levels in patients treated with AAV8 carrying *HEXA* and *HEXB* genes without an apparent reduction in the GM2 gangliosides species [16]. Nevertheless, the authors demonstrated an increased myelinization in the anterior and posterior corpus callosum [16]. Consequently, we hypothesize that our CRISPR/nCas9-based GT could have a similar or even better outcome than classical GT, and preclinical assays are ongoing on SD mice to test this notion.

Under our LP experimental conditions, the β-hexosaminidase activity levels allowed us to obtain a positive outcome in all the biomarkers evaluated, similar to previous results for MPS I and MPS IVA using CRISPR/Cas9 [24,38,39]. In this regard, Ou et al. (2020) reported CRISPR/Cas9-based edition assisted by the adeno-associated virus (AAV) 8 for GM2 gangliosidoses using a chimeric protein termed Hex M and the albumin locus as the integration site [21]. Hex M contains the α-subunit active site and the GM2AP-interaction portion of the β-subunit [22]. Although this strategy failed to reach a significant GM2 ganglioside reduction in the brain of SD animals, surprisingly, the authors reported a rescue in the rotarod analysis and a reduction in the neural vacuolation, suggesting a decrease in the accumulation of several substrates. Even though we did not measure the GM2 ganglioside, the findings on lysosomal mass and mtROS for LPs-assisted delivery agree with those of Ou et al. (2020) [21].

Even though viral vectors continue to be widely used as carriers in GT, there is a need to develop alternative strategies to overcome some of the current challenges of the virus-based approaches, such as random integration and activation of the adaptive immune response [40,41,42]. Consequently, after validating the CRISPR/nCas9 system in TSD and SD fibroblasts by LPs-assisted delivery, we decided to evaluate the suitability of an MLPs-based formulation. The results showed that the MLPs increased the transfection efficacy of the CRISPR/Cas9 system on TSD and SD fibroblasts compared to LPs, without inducing a significant reduction in cell viability (Figure 3). Nevertheless, we only achieved 1.9% and 2.2% of the wild-type levels for Hex A and Hex B, respectively, which were lower than those obtained with LPs (Figure 5). Since increased transfection upon MLPs treatment is generally linked to a higher CRISPR/Cas9-based genome edition, we put forward two major hypotheses to explain the discrepancies with the enzyme activity observations: (1) we cannot exclude any unknown harmful effect of the CRISPR/Cas9-based genome editing on the proliferation ratio of the edited cells, with respect to the unedited ones, and (2) the epigenetic regulation of CMV promoter causing a decrease in the transgene expression. Although both hypotheses are plausible and previously considered [43,44,45], we will need to perform further experiments to validate this premise under our experimental conditions.

Finally, we failed to identify any improvement in the GAGs accumulation or the lysosomal mass one month post-treatment with MLPs; however, we observed a significant recovery of the mtROS levels (Figure 7). Impaired oxidative stress control has been related to the major pathophysiological events in GM2-gangliosidoses [26]. We consider that despite the low β-hexosaminidase activity observed, it was enough to positively impact this key GM2 gangliosidoses biomarker; however, the exact mechanism remains to be elucidated.

## 4. Materials and Methods

Mammalian cell culture. Human embryonic kidney (HEK293) cells were purchased from ThermoFisher Scientific (Waltham, MA, USA) and maintained in Dulbecco’s Modified Eagle Medium (DMEM, Biowest, Lakewood Ranch, FL, USA) supplemented with 10% inactive fetal bovine serum (iFBS, Biowest, Lakewood Ranch, FL, USA), 100 U/mL penicillin, and 100 µg/mL streptomycin (1X P/S, Gibco, Waltham, MA, USA). The human Tay–Sachs (TSD, GM00515) and Sandhoff (SD, GM00317) fibroblasts were obtained from the Coriell Institute for Medical Research. Wild-type skin fibroblasts were isolated from a healthy adult man by informed consent. Fibroblasts were cultured in DMEM plus 15% FBS not inactivated, 1X P/S. The cells remained in a humidified atmosphere with 5% CO_2_ and 37 °C. Cells were used in passes between 10 to 20. All the experiments were approved by the Research and Ethics Committee of the Faculty of Science at Pontificia Universidad Javeriana (Minute 06, 2018).

CRISPR/nCas9 system and single guide RNA (sgRNA). Two sgRNAs previously engineered against the *AAVS1* locus [24] were cloned into an all-in-one CRISPR/nCas9 plasmid vector (Addgene: 74120, Watertown, MA, USA) through the golden gate strategy [46]. These sgRNAs showed no off-target effects on the *AAVS1* locus for the top 10 predicted sequences [24].

Design of Donor AAVS1 containing *HEXA* and *HEXB* cDNA. Donor AAVS1:Empty was previously designed, which includes two restriction enzymes for *Mfe*I and *Mlu*I flanking an expression cassette [24]. The human *HEXA* and *HEXB* cDNA sequences were a kind gift of Prof. Dr. Timothy Cox of the University of Cambridge, UK, and they were used as the template for PCR amplification. Specific primers were used to obtain the *HEXA* and *HEXB* cDNA with the overlapping *Mfe*I and *Mlu*I sequences at 5- and 3-end, respectively (Appendix A). The PCR products were cloned into a pJET 1.2 plasmid (Thermo Fisher Scientific, Waltham, MA, USA) according to the manufacturer’s indications. Sanger sequencing confirmed the positive clones carrying the full *HEXA* and *HEXB* sequences. Q5^®^ high fidelity polymerase (New England Biolabs, Ipswich, MA, USA) was used for all the PCR reactions. Through enzyme digestion with *Mfe*I-HF and *Mlu*I-HF enzymes (New England Biolabs, Ipswich, MA, USA), the *HEXA* and *HEXB* cDNA were extracted from pJET 1.2 and subcloned into Donor AAVS1:Empty. Henceforth, the resulting vectors were termed as Donor AAVS1:HEXA and Donor AAVS1:HEXB (Figure 1a).

Cell-penetrating bioconjugates and lecithin-based liposome synthesis. Magnetite nanoparticles (MNPs) were prepared by a chemical co-precipitation of FeCl_2_ and FeCl_3_, as described previously [47]. A redox reaction in the presence of Ag^+^ ions conducted by adding dropwise 5 M NaOH pH 8.5 to form a silver shell on the surface of MNPs. The solution was then stirred at 500 rpm and room temperature. A pH-responsive polymer poly(2-(dimethylamino)ethyl methacrylate) (pDMAEMA, Sigma, Burlington, MA, USA) was subsequently conjugated through a Hofmann reaction resulting in the MNPs:Ag-pDMAEMA nanoconjugates. The MNPs’ core was also functionalized with 3-aminopropyl)triethoxysilane (APTES, Sigma, Burlington, MA, USA) followed by conjugation of amino PEG12 propionic acid (PEG12, Sigma, Burlington, MA, USA). Finally, we immobilized the peptide Buforin II (BUF-II) to form the MNPs:Ag-pDMAEMA/PEG12-BUF-II nanobioconjugates henceforward denominated as MNPs@Ag-pD/BUF-II nanobioconjugates. We encourage readers to consult Ramírez-Acosta et al. (2020) for details [47]. Liposomes were synthesized by the lipid bilayer hydration method as described previously [48]. Briefly, 10 mg/mL of soy lecithin dissolved in chloroform was evaporated in a rotary evaporator (Hei-VAP Core, Heidolph, Germany) at 45 °C for one hour; 1X PBS was then added to hydrate the film at 55 °C for one additional hour. The solution was filtered through a 0.22 μm membrane and stored at 4 °C until further use.

Nanobioconjugates and liposome characterization. Hydrodynamic diameter and Z potential were determined using dynamic light scattering in a Zeta-Sizer Nano-ZS (Malvern Panalytical, Malvern, UK) as described previously [47]. To determine the maximum DNA loading charge of MNPs@Ag-pD/BUF-II nanobioconjugates, the AIO-mCherry plasmid (Addgene: 74120, Watertown, MA, USA) was mixed in amounts ranging from 0–3000 ng with 250 µg of the MNPs@Ag-pD/BUF-II nanobioconjugates as reported by Ramirez et al. (2020) [47]. Maximum DNA loading was defined as the ng DNA/µg MNPs@Ag-pD/BUF-II.

Loading CRISPR/nCas9-related plasmids on the MNPs@Ag-pD/BUF-II nanobioconjugates. An amount of 25 µg of the MNPs@Ag-pD/BUF-II nanobioconjugates were resuspended in 1X PBS pH 8.0 and precipitated with Merck Millipore PureProteome™ Magnetic Stands. After supernatant discard, 1 µg total pDNA in 1X Tris-Borate-EDTA (~0.5 µg pDNA CRISPR/nCas9 and/or ~0.5 µg pDNA Donor AAVS1:HEXA, Donor AAVS1:HEXB) was added. Accoupling reaction was allowed for 10 min at 130 rpm and room temperature. Finally, the MNPs@Ag-pD/BUF-II:pDNA complexes were mixed with 0.1 mg/mL liposomes dispersed in supplemented DMEM to obtain magnetoliposomes (MLPs).

### 4.1. Molecular Validation on HEK293 Cells

Homologous recombination (HR) assay. A total of 50,000 cells/well of HEK293 cells were seeded 24 h before experiments on 12-well plates. Later, cells were transfected with CRISPR/nCas9 and Donor AAVS1:HEXA or Donor AAVS1:HEXB at a ratio of 1:1 using Lipofectamine 3000 (LP, Thermo Fisher Scientific, Waltham, MA, USA) for 12 h. Next, the genomic DNA (gDNA) was isolated by a Monarch^®^ Genomic DNA Purification Kit (New England Biolabs, Ipswich, MA, USA). The gDNA was used as the template for a PCR, using primers to the *AAVS1* outside locus and the CMV sequence (Appendix A) [24]. The PCR product was purified with GeneJET Gel Extraction Kit (Thermo Fisher Scientific, Waltham, MA, USA) and sequenced by the Sanger method.

Transfection ratio. CRISPR/nCas9 and Donor AAVS1:Empy, Donor AAVS1:HEXA, or Donor AAVS1:HEXB were transfected with LPs on 80% confluent HEX293 cells plated on 12-wells plates (TPP; Trasadingen, Switzerland) at a 1:1 ratio. After 48 h, cells were harvested by trypsinization, washed twice with 1X phosphate buffer solution (1X PBS) and resuspended in 1X Hank’s Balanced Salt Solution (1X HBSS) supplemented with 5% FBS. CRISPR/nCas9 and Donor plasmids were analyzed by mCherry (*Exc/Emi*: 587/610) and enhanced green fluorescent protein (EGFP, *Exc/Emi*: 498/518 nm) expression, respectively in a BD FACSAria™ III Cell Sorter (Becton Dickinson, Franklin Lakes, NJ, USA). Propidium Iodide (1mg/mL PI, Sigma-Aldrich, St. Louis, MO, USA) was used to determine viable cells. A total of 50,000 events were acquired, and only single cells were included for analysis. All the flow cytometry data were analyzed with the FlowJo^®^ version 10.8 (Franklin Lakes, NJ, USA).

Transcription assays. Absolute quantitative real-time PCR (RT-qPCR) was performed to determine the *HEXA* and *HEXB* cDNA expression from the Donor AAVS1 plasmids. Briefly, 80,000 cells/well of HEK293 cells were transfected with 0.5 µg Donor AAVS1:HEXA or Donor AAVS1:HEXB plasmids using LPs for up to 24 h. Total RNA was extracted using the Monarch^®^ Total RNA Miniprep Kit (New England Biolabs, Ipswich, MA, USA), checked by a denaturing agarose gel [49], and quantified by a NanoDrop 1000 Spectrophotometer (Thermo Fisher Scientific, Waltham, MA, USA). An amount of 0.5 µg of RNA was retrotranscribed with a High-Capacity cDNA Reverse Transcription Kit (Applied Biosystems, Waltham, MA, USA), according to the manufacturer’s instructions. The resulting cDNA was then used for real-time PCR. Specific TaqMan probes for human *HEXA* (Hs00942655_m1-FAM) and *HEXB* (Hs01077594_m1-FAM) were purchased from Applied Biosystems (Thermo Fisher Scientific, Waltham, MA, USA). Finally, the number of copies/μL was calculated from a standard curve. Untransfected cells were used to determine the basal expression for each gene.

β-hexosaminidase activity. The β-hexosaminidase activity was performed following the protocols described by Tropak et al. (2004) and Mauri et al. (2013) with slight modifications [50,51]. Briefly, monolayer cells plated on 12-well plates (TPP, Trasadingen, Switzerland) were washed twice with 1X PBS, and then 300 µL of citrate–phosphate buffer (CP buffer; pH 4.2) containing 0.5% Triton X-100 was added. After 30 min, the supernatant was collected and clarified by centrifugation at 4000 rpm/10 min. An amount of 50 µL of supernatant was pipetted into 96-well black flat-bottom microplates (Corning, NY, USA) and incubated with 20 µL of 4-methylumbelliferyl-β-d-acetyl-glucosaminide sulfate (MUGS 3.2 µM, Calbiochem, San Diego, CA, USA), or 4-methylumbelliferyl-β-d-acetyl-glucosaminide (MUG 3.2 µM, Sigma-Aldrich, St. Louis, MO, USA) for measuring the Hex A and Hex B activity, respectively [10]. For Hex B activity determination, the samples were incubated at 52 °C for 2 h before incubation with the MUG substrate [11]. After 20 min of reaction at 37 °C, 150 µL of stop buffer (0.2 M Glycine, pH 10.8) was added. The plate was read in a spectrofluorometer Twinkle LB970 (*Exc/Emi*: 365/450 nm; Berthold Technologies, Bad Wildbad, Germany). The protein concentration was quantified using a BCA Protein Assay Kit (Thermo Fisher Scientific, Waltham, MA, USA). One unit of enzyme activity (U) was defined as the amount of enzyme catalyzing the hydrolysis of 1 nmol of substrate per hour. A standard curve of 4-methylumbelliferone (Sigma-Aldrich, St. Louis, MO, USA) was built to determine the relative units of fluorescence associated with 1 nmol. Specific Hexosaminidase activity was expressed as U/mg.

### 4.2. Delivery of the CRISPR/nCas9 System and Donor AAVS1 to GM2 Fibroblasts

Viability assays. To determine the effect of the MNPs@Ag-pD/BUF-II on the viability of GM2 fibroblasts, we performed preliminary assays with MTT (3-(4,5-dimethylthiazol-2-yl)-2,5-diphenyltetrazolium bromide; Sigma-Aldrich, St. Louis, MO, USA) and an LDH Cytotoxicity Detection Kit (Lactate dehydrogenase release, Sigma, Burlington, MA, USA). Briefly, 10,000 cells/well seeded on 96-well plates (TPP, Trasadingen, Switzerland) were incubated with a 25 µg/mL/0.05 mg/mL MNPs@Ag-pD/BUF-II:liposome ratio for up to 48 h. A microplate reader (Biochrom Anthos 2020, Cambridge, UK) was used to read the microplates at 562/630 nm and 490/630 nm for MTT and LDH, respectively. Moreover, apoptosis assays were conducted with the aid of an Alexa FluorTM 488 Annexin/Dead Cell Apoptosis Kit (ThermoFisher Scientific, Waltham, MA, USA). For instance, 30,000 cells/well growing on 12-well plates (TPP, Trasadingen, Switzerland) were incubated under the conditions described. After 24 h of exposure to treatments, cells were harvested by trypsinization and processed by flow cytometry. A total of 50,000 events were recorded in a BD FACSAria™ III Cell Sorter (Becton Dickinson, Franklin Lakes, NJ, USA). The obtained data were analyzed with the aid of the FlowJo^®^ software (Franklin Lakes, NJ, USA). Only viable single cells were included in the analysis. Fibroblasts incubated with 2.5 µM doxorubicin for 72 h were included as a positive control of apoptosis. Annexin V-Alexa Fluor 488 (*Exc*/*Emi*: 488/530 nm) positive cells were interpreted as early apoptotic cells, while propidium iodide (PI, *Exc*/*Emi*: 488/695 nm) positive cells as necrotic ones. Annexin^+^/PI^+^ cells were identified as late apoptotic cells. Finally, the cells were observed with an Axio Observer Z1 microscope (ZEISS, Birkerød, Denmark) by differential interference contrast (DIC). The images were analyzed with the aid of the ImageJ^®^ version 1.53t (NIH, Washington, DC, USA) [52].

Internalization assays. To evaluate the internalization of MLPs containing Rhodamine-labeled MNPs@Ag-pD/BUF-II (Rho-MLPs) on GM2 fibroblasts, 30,000 cells/well were seeded on PolyD-Lysine covered coverslips. The fibroblasts were incubated at 4 °C or 37 °C for four hours. After one wash with 1X PBS, the cells were fixed with 4% paraformaldehyde in PBS (4% PFA) at room temperature. Finally, the coverslips were put on slides containing a drop of fluoroshield mounting medium with DAPI (Abcam; Cambridge, UK). The cells were observed with an Axio Observer Z1 microscope (ZEISS, Birkerød, Denmark). All the images were analyzed with the aid of the NIH ImageJ^®^ version 1.53t (Washington, DC, USA) [52].

Transfection ratio. To evaluate if an increase in the ratio transfection efficiency could be achieved with MLPs in comparison with LPs, we performed transfection ratio experiments. In brief, GM2 fibroblasts were seeded on 12-well plates (TPP, Trasadingen, Switzerland) and transfected with LPs and MLPs carrying Donor AAVS1:HEXA or Donor AAVS1:HEXB. At 48 h post-interaction, cells were harvested and analyzed by EGFP expression using flow cytometry (*Exc/Emi*: 498/518 nm). Cells were then analyzed as described above for the HEK293 cells.

Transfection assays. A total of 30,000 cells were seeded on 12-well plates (TPP; Trasadingen, Switzerland) 24 h before the transfection assays. One hour before the assays, the medium was replaced by a fresh one. Cells were then transfected with Lipofectamine 3000 (Thermo Fisher Scientific, Waltham, MA, USA) following the manufacturer’s instructions. For MLPs, a 25 µg/mL/0.05 mg/mL MNPs@Ag-pD/BUF-II:liposome ratio was added dropwise on cells seeded in 12-well plates (TPP; Trasadingen, Switzerland). An amount of 1µg of DNA was used in all the experiments with LPs and MLPs. CRISPR/nCas9 and Donor AAVS1 plasmids were used at a 1:1 ratio. Cells were monitored for up to 1 month. The transcription and enzyme activity were performed as described above for the HEK293 cells. Naked plasmid DNA was included as control with a non-significant impact on any of the biological assays conducted here.

Glycosaminoglycans determination. As secondary storage of GAGs can occur in GM2 gangliosides [2,26,53], we decided to evaluate the impact of the CRISPR/nCas9 on the GAGs accumulation following a protocol described previously [24,25]. For one month, every four days, 300 µL of culture media were collected and stored at −20 °C. The samples were processed as a unique sample pool. Then, 50 µL were mixed with 275 µL of 1% p/v 1,9-dimethyl methylene blue (DMB, Sigma-Aldrich, St. Louis, MO, USA) pH 3.3 dissolved in 2M Tris-Base (Sigma-Aldrich, St. Louis, MO, USA) for 1 min at room temperature. The samples were read for the first 5 min using a spectrophotometer BioSpec-1601 (Shimadzu, Canby, OR, USA) at 520 nm. A standard curve was built using chondroitin sulfate A (Sigma-Aldrich, St. Louis, MO, USA).

Lysosomal mass evaluation. The lysosomal staining was performed using LysoTracker^TM^ Deep Red (*Exc*/*Emi*: 647/668 nm; Thermo Fisher Scientific, Waltham, MA, USA) as described previously [24]. Briefly, after one month of treatment with the CRISPR/nCas9 system assisted by either LPs or MLPs, cells were stained with 50 nM or 75 nM of LysoTracker^TM^ for flow cytometry and fluorescence microscopy, respectively. Cells were then incubated at 37 °C/5% CO_2_ for one hour and harvested by trypsinization for flow cytometry experiments. Then, the cells were homogenized in 1X HBSS and their viability was evaluated by PI. A total of 50,000 events were recorded in a BD FACSAria™ III Cell Sorter. Viable single cells were analyzed in the FlowJo^®^ software (Franklin Lakes, NJ, USA). For microscopy assays, cells were washed twice after staining with LysoTracker^TM^, fixed with 4% PFA, and placed on slides containing one drop of Fluoroshield mounting medium (Abcam, Cambridge, UK). The slides were imaged with an Axio Observer Z1 microscope (ZEISS, Birkerød, Denmark) and analyzed using ImageJ^®^ software (NIH, Washington, DC, USA) [52].

Mitochondrial-related oxidative stress (mtROS). To evaluate the impact of the CRISPR/nCas9-based system on the mtROS, after one month of treatment, cells were incubated with 2.5 µM MitoSOXTM Red indicator (Thermo Fisher Scientific, Waltham, MA, USA) in Hank’s Balanced Salt Solution (1X) for 15 min. Then, cells were harvested, washed twice with 1X PBS, and passed through a Cytek^®^ Aurora CS cytometer (Cytek, Fremont, CA, USA) using PE—Red (*Exc/Emi*: 510/580 nm) channel. Positive controls were conducted by incubating the fibroblasts in the presence of 100 µM cobalt chloride (CoCl_2_; Sigma-Aldrich, St. Louis, MO, USA) [24].

Griess assay. Nitric oxide-derived species were determined using the Griess reagent (Sigma-Aldrich, St. Louis, MO, USA) [54]. An amount of 100 µL of supernatant from long-term treated or untreated cells was mixed with 100 µL of the reagent and incubated for 15 min at room temperature on 96-well clear flat bottom microplates (Corning, Steuben County, NY, USA). Then, the plate was read using a microplate reader (Biochrom Anthos 2020, Cambridge, UK) at 562/630 nm. Fibroblasts incubated with 1 µM lipopolysaccharide *Escherichia coli* O111:B4 (Sigma-Aldrich, St. Louis, MO, USA) were included as the positive control. A standard curve of sodium nitrite (NaNO_2_, Sigma-Aldrich, St. Louis, MO, USA) was built to compare the data obtained from long-term treated or control cells.

Statistical analysis. Experimental data were analyzed in GraphPad Prism^®^ version 8.0.0 for Mac (GraphPad Software, San Diego, CA, USA). The data is presented as mean ± standard error (SE). Shapiro–Wilk Test and Leve Test were used to evaluate the normal distribution and homoscedasticity. Mean comparison between groups was made with Student’s *t*-test, Mann–Whitney U test, or ANOVA, according to the normality test findings; *p* < 0.05 was considered statistically significant.

## 5. Conclusions

In this work, we demonstrated that our previously developed CRISPR/nCas-based genome editing could be used to insert the *HEXA* and *HEXB* genes cloned in the Donor AAVS1:Empty into the *AAVS1* locus. In addition, we observed a significant increase in the intracellular Hex A and Hex B enzyme activity, using conventional LPs as the carrier, which led to the recovery of critical GM2 gangliosides biomarkers, such as the secondary GAGs accumulation, the lysosomal mass, and the mtROS reduction. In contrast, for MLPs formed by plasmid-loaded MNPs@Ag-pD/BUF-II nanobioconjugates, we also achieved a slight increase in the intracellular β-hexosaminidase activity, which had a positive outcome in the recovery of the mtROS levels in both TSD and SD fibroblasts. Preclinical assays are currently ongoing on SD mice to determine the potential effect of our CRISPR/nCas9-based gene therapy strategy prior to moving to clinical trials.

## Figures and Tables

**Figure 1 ijms-23-10672-f001:**
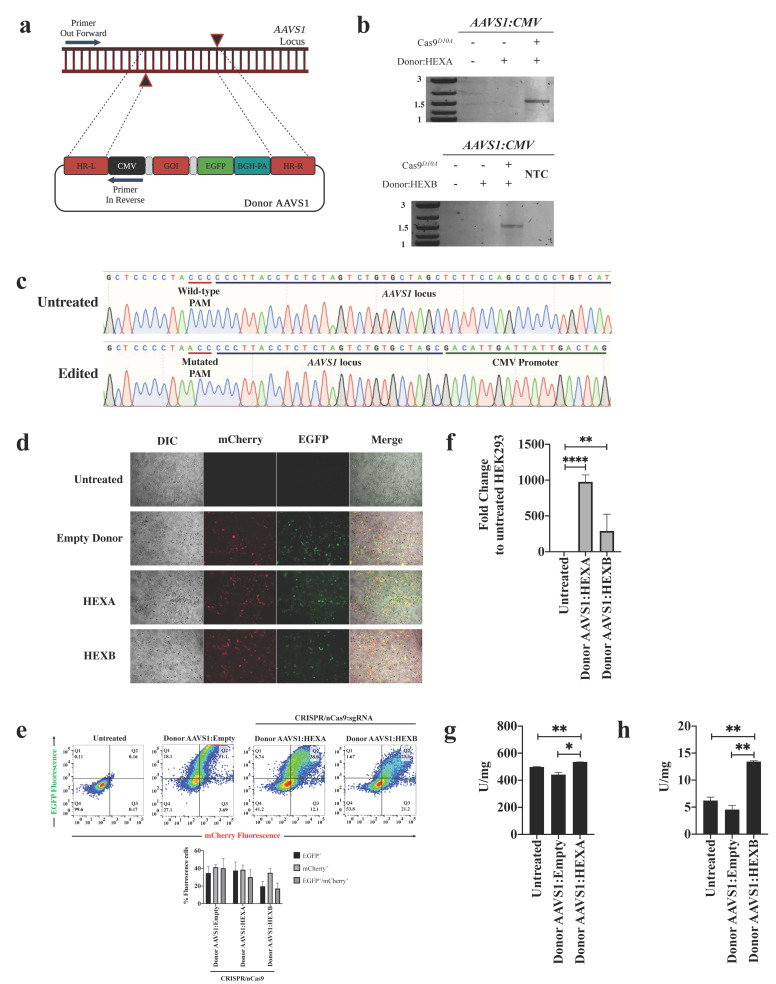
Validation of the CRISPR/nCas9 system on HEK293 cells. (**a**) Molecular strategy for homologous recombination assays. A set of primers flanking a sequence of the AAVS1 locus (which is absent in the homologous recombination left arm) and the CMV promoter were designed to amplify a region of 1.5 kb. Only the successful knock-in of the Donor AAVS1 allows for amplifying the expected fragment. GOI refers to HEXA or HEXB cDNA. (**b**) PCR products from cells transfected with CRISPR/nCas9 and Donor AAV1:HEXA or Donor AAV1:HEXB plasmids. (**c**) Electropherograms from Sanger sequencing performed on PCR products. In the reverse sequence, note the mutated PAM sequences inserted in the Donor plasmid [24]. (**d**) Epifluorescence assays to visualize the EGFP and mCherry signals. (**e**) Transfection ratio quantitation by flow cytometry (*n* = 5). Transcription assays ((**f**), *n* = 3) and specific Hex A ((**g**), *n* = 4) or Hex B ((**h**), *n* = 4) enzyme activities are also shown. * *p* ≤ 0.05, ** *p ≤* 0.01, **** *p ≤* 0.0001. Two-tailed Student’s *t*-test.

**Figure 2 ijms-23-10672-f002:**
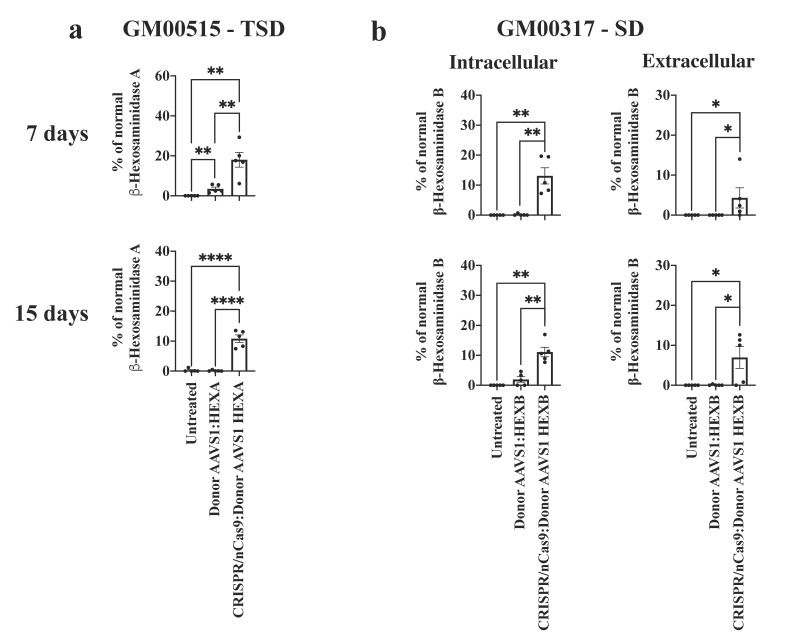
Short- and medium-term genome editing on GM2 fibroblasts. (**a**) Intracellular Hex A enzyme activity for TSD fibroblasts 7 and 15 days post-edition (*n* = 4). (**b**) For SD fibroblasts, the intra- and extracellular Hex B activity are shown in the left and right panels, respectively (*n* = 4). * *p* ≤ 0.05, ** *p* ≤ 0.01, **** *p* ≤ 0.0001. Mann–Whitney U test.

**Figure 3 ijms-23-10672-f003:**
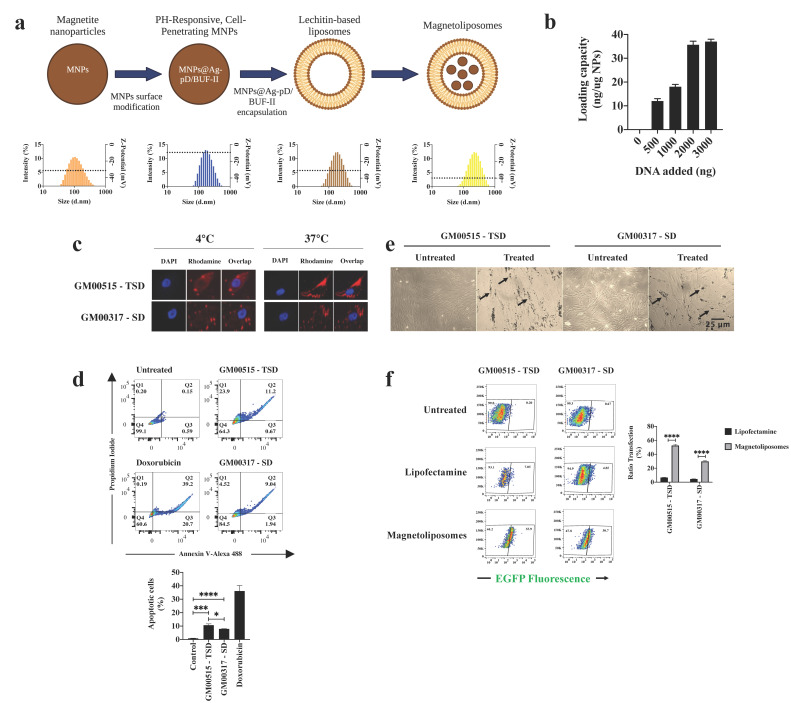
Cell viability and transfection ratio of MLPs on GM2 fibroblasts. (**a**) Upper panel shows the experimental strategy for MLPs synthesis, while the bottom one shows the hydrodynamic diameter and Z-potential (dotted line). (**b**) Loading DNA capacity. (**c**) MLP internalization assays. Note the intracellular accumulation of the Rho-labeled MLPs when the GM2 fibroblasts are incubated at 37 °C, while their incubation at 4 °C resulted in a peripheral distribution. (**d**) Apoptosis assays performed on GM2 fibroblasts. The upper panel shows a representative trace for each quadrant. Q2 quadrant was interpreted as late apoptotic cells. The lower panel shows the mean for three independent experiments. GM00515-TSD fibroblasts were more susceptible to interaction with MLPs than GM00317-SD fibroblasts. (**e**) Light microscopy of GM2 fibroblasts after their incubation with MLPs. Black arrows point to MNPs@Ag-pD/BUF-II nanobioconjugates. (**f**) Ratio transfection assays. A representative cytogram is shown in the left panel, while the mean of three experiments is presented in the right one. * *p* ≤ 0.05, *** *p* ≤ 0.001, **** *p* ≤ 0.0001. Two-tailed Student’s *t*-test.

**Figure 4 ijms-23-10672-f004:**
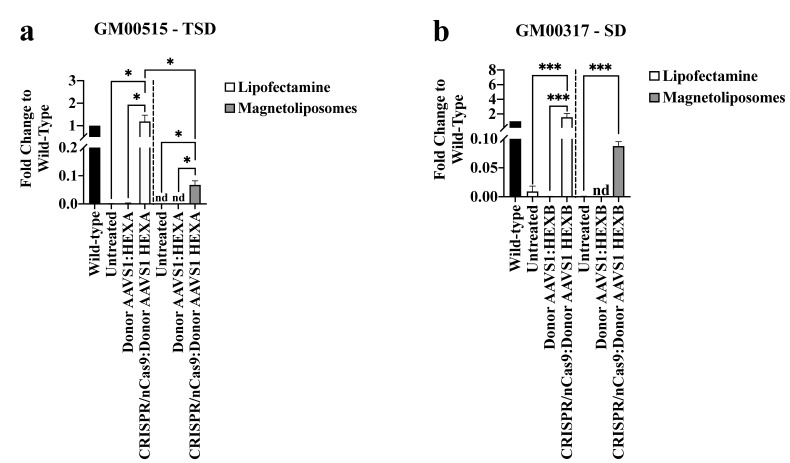
Transcription assays on long-term edited GM2 fibroblasts. The fold-change in the mRNA levels for GM00515-TSD (**a**) and GM00317-SD (**b**) fibroblasts one-month post-treatment are shown (*n* = 3). * *p* ≤ 0.05, *** *p* ≤ 0.001. GM00515-TSD and GM00317-SD were analyzed using the two-tailed Student’s *t*-test and Mann–Whitney U test, respectively.

**Figure 5 ijms-23-10672-f005:**
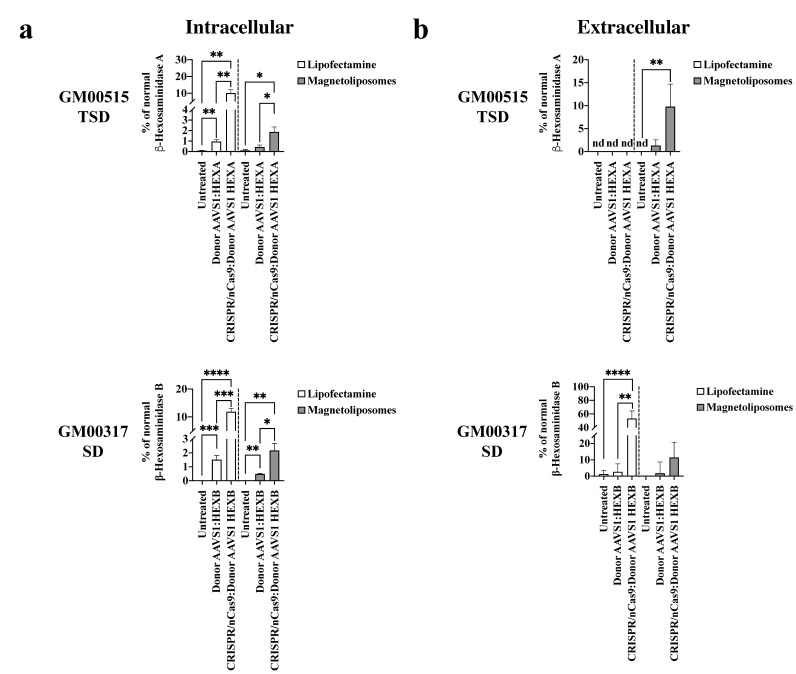
β-hexosaminidase activity on long-term edited GM2 fibroblasts. The intracellular (**a**) and extracellular (**b**) enzyme activity to wild-type levels for GM00515-TSD and GM00317-SD fibroblasts is shown (*n* = 4). * *p* ≤ 0.05, ** *p* ≤ 0.01,*** *p* ≤ 0.001, **** *p* ≤ 0.0001. Two-tailed Student’s *t*-test.

**Figure 6 ijms-23-10672-f006:**
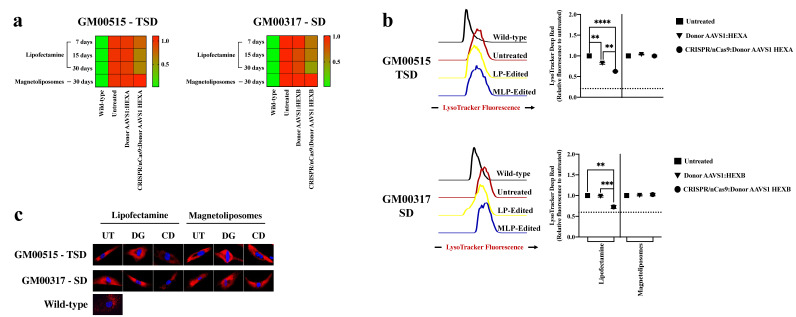
GAGs and lysosomal mass determination on long-term edited GM2 fibroblasts. (**a**) A heat map shows the changes in the GAGs accumulation in both GM00515-TSD and GM00317-SD fibroblasts after one month of treatment (*n* = 4). (**b**) Lysosomal mass was quantified by flow cytometry. The left panel shows a representative histogram for each experimental group. The right panel shows the mean of four independent experiments. (**c**) Epifluorescence microscopy images at 60× for GM00515-TSD and GM00317-SD fibroblasts stained for lysosomal mass analysis. Blue fluorescence is for the nuclei. UT: Untreated. DG: Donor AAVS1:GOI, CD: CRISPR/nCas9 plus Donor AAVS1:GOI. ** *p* ≤ 0.01, *** *p* ≤ 0.001, **** *p* ≤ 0.0001. Two-tailed Student’s *t*-test.

**Figure 7 ijms-23-10672-f007:**
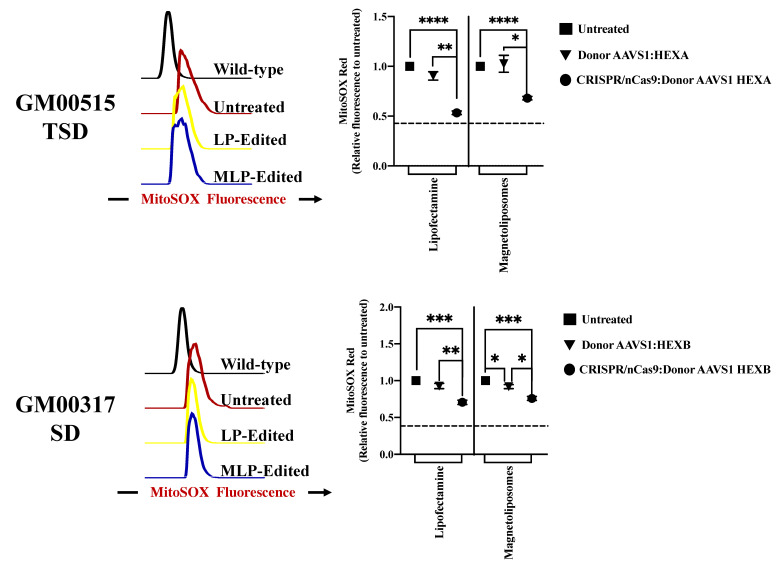
mtROS response on long-term edited GM2 fibroblasts. MitoSOX red fluorescence intensity for GM00515-TSD and GM00317-SD fibroblasts after one month of treatment. The left panels show representative histograms, while the right panels show the mean of three independent experiments. * *p* ≤ 0.05, ** *p* ≤ 0.01, *** *p* ≤ 0.001, **** *p* ≤ 0.0001. Two-tailed Student’s *t*-test.

## Data Availability

Raw data will be available upon reasonable request.

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
