# Peer review of "CRISPR/nCas9-Based Genome Editing on GM2 Gangliosidoses Fibroblasts via Non-Viral Vectors"

_ijms, 2022, doi:10.3390/ijms231810672_

Round 1

Reviewer 1 Report

In this manuscript, the investigators use fibroblast cell culture models of Tay-Sachs or Sandhoff Disease to test the feasibility of correcting Lysosomal Storage Genetic Disorders, which are the result of insufficient enzyme activities.  There currently are no satisfactory treatments, with enzyme replacement therapies providing a modest benefit.  Some investigators are testing viral therapies, using AAV8 e.g. which has a “safe harbor” chromosomal integration site (AAVS1), but the authors correctly point out that there are limitations and alternate avenues of investigation are needed.  Thus, this is the second of two papers in which the authors attempt to optimize the introduction of a cassette harboring an exogenous promoter and either GALNS (previous paper, Leal, A.F. and C.J. Alméciga-Díaz, Efficient CRISPR/Cas9 nickase-mediated genome editing in an in vitro model of mucopolysaccharidosis IVA. Gene Ther, 2022),  or HEXA and HEX B coding sequences (current manuscript).  Homologous integration into AAVS1, as well as upregulation of HEX RNA and activity can be observed in Cas9 nickase + sgRNAs + HEX sequences but none of the controls.

The main question is what is innovative besides the addition of two additional LSD models?  One thing is the use of different methods for transfection including a comparison of previously used lipofection vs. different techniques using magnetic nanoparticles and modifying them making them PH-responsive, and enclosing them in lecithin-based liposomes. The MNPs appear to have a higher transfection efficiency than lipofectamine (Fig 3), although the latter appears to fare better in the long term.  In addition, lysosomal mass is reduced to a greater extent in lipofectamine-transduced vs. MLP- assisted.  However, mtROS stress, which may be one pathway for LDS pathogenesis is reduced, and thus preclinical models are proposed for further study.  I believe that the authors are realistic in their overall discussion of the strengths and limitations of their study.  Because of this, and the importance of building on different  approaches to LSD therapies, I would recommend publication after minor modifications.

Minor weaknesses:

Lines 111-112 Can be rewritten e.g. “activity increased significantly 7 and 15 days after … editing …” (sounds like treatment was for 15 days, and can use editing or synonym, or use edition as a noun). 

116 observed IN supernatants

120 Same as above (111) 7 and 15 days post-treatment or 15 days after treatment (same reason)

122 editing

121 suggests long term (delete “a”)

537 Insert HEX into the AAVS1 locus

Author Response

Reviewer 1

Lines 111-112 Can be rewritten e.g. “activity increased significantly 7 and 15 days after …editing…” (sounds like treatment was for 15 days, and can use editing or synonym, or use edition as a noun).

Answer. We appreciate this comment. We have included post-edition to clarify this point. Please see line 117.

116 observed IN supernatants

Answer. We have corrected this typing issue. See line 121.

120 Same as above (111) 7 and 15 days post-treatment or 15 days after treatment (same reason).

Answer. We have fixed this sentence. Please see line 125.

122 editing.

Answer. We have fixed this word. Please see line 127.

121 suggests long term (delete “a”).

Answer. We have deleted it.

537 Insert HEX into the AAVS1 locus.

Answer. We have fixed the sentence. Please see lines 569-570.

Reviewer 2 Report

Andrés Felipe Leal and co-authors present a quality and well-written experimental manuscript describing CRISPR/nCas9-based genome editing on GM2 gangliosidoses fibroblasts via non-viral vectors.

Authors validated a CRISPR/Cas9-based gene editing strategy that relies on a Cas9 nickase (nCas9) as a potential approach for treating GM2 gangliosidoses using in vitro models for TSD and SD. They employed nCas9 contains a mutation in the catalytic RuvC domain but maintains the active HNH domain, which reduces potential Off-target effects. Classical liposome (LP)- and novel magnetoliposomes (MLPs)-based were used as vectors to deliver the CRISPR/Cas9 system. When LP was used as a vector, positive outcomes were observed for the β-hexosaminidase activity, glycosaminoglycans levels, lysosome mass, and oxidative stress. In the case of MLPs, a high cytocompatibility and transfection ratio was observed, with a slight increase in the β-hexosaminidase activity and significant oxidative stress recovery in both TSD and SD cells. These results show the remarkable potential of CRISPR/nCas9 as new alternative for treating GM2 gangliosidoses, as well as the superior performance of non-viral vector to enhance the potency of this therapeutic approach. 

Authors evaluated CRISPR/nCas9-based gene-editing system to induce the insertion of normal HEXA and HEXB cDNA into the AAVS1 locus on in vitro models of TSD and SD, respectively. After initial validation experiments, they extended the findings to long-term experiments using two delivery strategies based on both classical lipo-transfection and a novel magnetolipo-transfection methods.

Finally, authors conclude that they failed to identify any improvement in the GAGs accumulation or the lysosomal mass after one-month post-treatment with MLPs; however, they observed a significant recovery of the mtROS levels. Impaired oxidative stress control has been related to the major pathophysiological events in GM2-gangliosidoses. Authors consider that despite the low β-hexosaminidase activity observed, it was enough to impact posi tively this key GM2 gangliosidoses biomarker; however, the exact mechanism remains to be elucidated.

Overall, the manuscript is highly valuable for the scientific community and should be accepted for publication after the corrections are made.

==============================

Other comments:

1) Please check for typos throughout the manuscript.

2) With regards to gene editing authors are kindly encouraged to cite the following article that describes the use of CRISPR/Cas9 system for therapeutic gene editing of certain transcription regulators. DOI: 10.3390/genes11060704

Author Response

Reviewer 2

1) Please check for typos throughout the manuscript.

Answer. We did complete review of the English grammar and typos.

2) With regards to gene editing authors are kindly encouraged to cite the following article that describes the use of CRISPR/Cas9 system for therapeutic gene editing of certain transcription regulators. DOI: 10.3390/genes11060704

Answer. We appreciate the reviewer’s comment. We have included that suggested paper in the introduction. Please see lines 60-61.

Reviewer 3 Report

In the current manuscript, authors have presented CRISPR/nCas9-based genome editing on GM2 gangliosidoses fibroblasts via non-viral vectors – liposome and magnetoliposomes.

The article is well structured into section and subsections. English is clear and professional. It is within the scope of journal.

There are some comments to improve the article:

1)     Page 6, Figure 3: The representation of statistical significance is missing in the lower panel of 3d and similarly in the right panel of 3f.

2)     Page 2, line 76: Is the subheading 2.1.1. missing? If not, the numbering needs to be corrected.

3)     Page 17-19: References – The referencing format needs to be consistent and complete. Some have missing volume and page numbers. Check reference number 1, 2, 6, 19, 21, 38, and 44. Also the reference number 13 has DOI information, which is not present in other references.

Author Response

Reviewer 3

1) Page 6, Figure 3: The representation of statistical significance is missing in the lower panel of 3d and similarly in the right panel of 3f.

Answer. We have reviewed all the manuscript and include all the statical symbols that were missing.

2) Page 2, line 76: Is the subheading 2.1.1. missing? If not, the numbering needs to be corrected.

Answer. We reviewed all the subheadings along the manuscript.

3) Page 17-19: References – The referencing format needs to be consistent and complete. Some have missing volume and page numbers. Check reference number 1, 2, 6, 19, 21, 38, and 44. Also the reference number 13 has DOI information, which is not present in other references.

Answer. We have reviewed the full document and fixed the references with a consistent format throughout the manuscript.